

# Study on the absorption characteristics of euscaphic acid and tiliroside in fruits of *Rosa laxa* Retz.

Ning Wang[1,2,3,4,5] and Li Tian[5]

[1] Department of Medicine, Dalian University of Technology, Dalian, Liaoning, China
[2] Liaoning Cancer Hospital & Institute, Shenyang, Shenyang, China
[3] Cancer Hospital of Dalian University of Technology, Shenyang, China
[4] Zhongshan Institute for Drug Discovery, Shanghai Institute of Materia Medica, Zhongshan City, China
[5] Chinese University of Traditional Chinese Medicine, Xinjiang Medical University, Urumqi, China

Corresponding authors
Ning Wang, wangning@zidd.ac.cn
Li Tian, tianli109@126.com

## ABSTRACT

The fruits of *Rosa laxa* Retz. (FRL) have a long history of medicinal use, known for their rich composition of flavonoids, polyphenols, amino acids, sugars, and other bioactive compounds. FRL exhibits pharmacological effects such as antioxidant, antiviral, antibacterial, and antitumor activities, making it a valuable resource with significant development potential in both the food and pharmaceutical industries. This study employed a response surface methodology combined with ultra-high-performance liquid chromatography-triple quadrupole mass spectrometry (UPLC-TQ-MS) to optimize FRL extraction. Reflux extraction was determined to be the most effective method with the following optimized parameters: 65% ethanol extraction solvent, material-to-liquid ratio of 1:35 (g/mL), and extraction time of 140 min, resulting in the FRL extract (FRLE). Under these optimized conditions, the extracted amount was extract was 51.00 ± 1.07%, the average content of total polyphenols was 126.55 ± 2.61 mg/g, and the average content of euscaphic acid was 2.90 ± 0.08 mg/g, demonstrating the efficiency of the extraction method. Using the Caco-2 cell model, the study investigated the absorption characteristics of euscaphic acid and tiliroside within FRLE. Results indicated that with increasing time, the absorbed amount ($Qr$) of euscaphic acid and tiliroside gradually increased, with an efflux ratio ($R_{B \to A / A \to B}$) of less than 1.5, suggesting bidirectional drug transport with no significant directionality. Upon the addition of P-glycoprotein (P-gp) inhibitors Verapamil (Ver) and Ciclosporin A (CsA), as well as the chelating agent ethylenebis (oxyethylenenitrilo) tetraacetic acid (EGTA), $Qr$ and $Papp$ values notably increased, indicating that these two components are P-gp substrates with cellular basolateral efflux transport. Additionally, optimal absorption efficiency was observed under weakly acidic conditions (pH 6.0). In conclusion, euscaphic acid and tiliroside in FRLE demonstrated good membrane permeability, primarily relying on passive diffusion for absorption. This study offers experimental insights into the intestinal absorption of FRL *in vivo*.

# INTRODUCTION

There are over 200 species of *Rosa* sp. worldwide, many of which produce fruit extensively used for both medicinal and culinary purposes (*Yang et al., 2020*). The rich bioactive components in these fruits, such as amino acids, polyphenols, organic acids, and sugars, contribute to their diverse pharmacological properties, including antioxidant, immunomodulatory, lipid-lowering, anticancer, and antibacterial activities (*Ayati et al., 2018*; *Chen et al., 2017*; *Nađpal et al., 2018*; *Phetcharat, Wongsuphasawat & Winther, 2015*). *Rosa laxa* Retz. is primarily found in central Siberia and the Altai Mountain region, typically growing in shrubs, dry ditches, and river valleys at altitudes ranging from 500 to 1,200 m (*Guantario et al., 2023*). This plant exhibits remarkable resilience to drought, infertile soil, and extreme cold, making it valuable for ecological restoration and enhancement (*Zaslavskaya & Malafiy, 2021*). The fruit of *Rosa laxa* Retz. (FRL) contains a range of nutritional components, including amino acids, sugars, minerals, and vitamins. It can be processed into juice or vitamin concentrate. Additionally, FRL demonstrates various physiological effects, such as antioxidation, antiviral, antibacterial, and lipid-lowering properties, all with minimal toxicity (*Cavalera et al., 2017*; *Nađpal et al., 2018*; *Phetcharat, Wongsuphasawat & Winther, 2015*). Therefore, FRL serves not only as a medicinal and edible resource but also as an environmentally friendly plant with substantial development potential. Research findings suggest that euscaphic acid and tiliroside found in FRL possess antioxidant and anticancer pharmacological activities and are characterized as non-toxic and non-mutagenic (*Dai et al., 2019*; *Jeong et al., 2022*; *Velagapudi, Aderogba & Olajide, 2014*; *Xu et al., 2024*). Given the therapeutic potential of FRL components, a thorough evaluation and optimization of extraction techniques for these bioactive compounds is essential. The use of response surface methodology (RSM) is particularly advantageous in elucidating the complex interactions between evaluation parameters and individual factors (*Zhang et al., 2022*). This approach enables the efficient identification of optimal extraction conditions, accounting for the influence of multiple variables, and is especially relevant for studies focused on extraction processes in herbal medicine research (*Addo et al., 2022*). Intestinal absorption represents a pivotal stage for drug assimilation into the human body. Scrutinizing the mechanisms governing drug absorption in the intestines is instrumental in comprehending the fundamental processes and molecular underpinnings of drug assimilation (*Awortwe, Fasinu & Rosenkranz, 2014*). The Caco-2 monolayer, derived from a human colon adenocarcinoma cell line, exhibits morphological, transport, and metabolic characteristics akin to small intestine cells, closely mimicking the mechanisms involved in intestinal absorption. Consequently, it is frequently employed for *in vitro* investigations into drug absorption kinetics and characteristics (*Awortwe, Fasinu & Rosenkranz, 2014*). In light of these considerations, the present study optimizes the extraction parameters for FRL using response surface

methodology, yielding the FRL extract (FRLE). Subsequently, leveraging the Caco-2 cell model, we delve into the absorption profiles of euscaphic acid and tiliroside within FRLE. This foundational research sets the stage for further exploration and development of FRL.

## MATERIALS AND METHODS

### Materials and reagents used in the experiment

FRL was collected in August 2019 from Yamalik Mountain in Urumqi city (87°54′E, 43°79′N; altitude, 1,120.0 m) with a temperature of 27 °C. It was identified by Professor Xu Haiyan of Xinjiang Medical University as the dried fruit of *Rosa laxa* Retz., a plant from the *Rosaceae* family. The Caco-2 cell line was purchased from Wuhan Boster Biological Engineering Co., Ltd (item number: CX0064; Wuhan, China). The passage numbers of the cells used in the experiments were between 20 and 40 (product number: CX0064). Euscaphic acid (YRY097-210101, ≥98%) and gallic acid (YRM048, ≥98%) were purchased from Chengdu Yijie Rui Biotechnology Co., Ltd (Chengdu, China). Transwell® 12-well transport chambers (5662), 12-well plates, 96-well plates, and culture flasks were all purchased from the Costar Corporation (Corning, NY, USA). Fetal bovine serum (FBS) (FND622) was purchased from EKOSAI Biotechnology Co., Ltd (Shanghai, China). Dulbecco's Modified Eagle Medium (DMEM) culture medium (Batch No.: 7121572), non-essential amino acids (11140050), penicillin-streptomycin dual-antibiotic solution (10378016), trypsin (25200056), and D-Hank's balanced salt solution (88284) were all purchased from the Gibco (Waltham, MA, USA). Fluorescein sodium (F8140), Verapamil (IV0040), Cyclosporin A (SC5120), ethylenebis (oxyethylenenitrilo) tetraacetic acid (EGTA) (E8050), and sterile dimethyl sulfoxide (DMSO) (D8371) were all purchased from Solarbio Technology Co., Ltd (Beijing, China). Methanol and acetonitrile (high-performance liquid chromatography (HPLC) grade) were purchased from the Thermo Fisher Scientific (Waltham, MA, USA).

### Establishment of the determination method for euscaphic acid and tiliroside

Prepare euscaphic acid and tiliroside to make a stock solution of 100 μg/mL, stored for future use (in the subsequent addition of the drug to the cell culture solution, the amount of DMSO should not exceed 0.1% of the volume of the cell culture solution).

#### *Chromatographic conditions*

The analysis was performed using a Waters (Milford, MA, USA) ACQUITY ultra-high performance liquid chromatography (UPLC) system with an HSS T3 column (100 mm × 2.1 mm, 1.8 μm). The column temperature was maintained at 30 °C, and the sample chamber was kept at 4 °C. The mobile phase consisted of Phase A (10 mmol/L ammonium acetate) and Phase B (acetonitrile), with gradient elution as follows: 0–1 min, 97% A; 1–3 min, 97% A → 30% A; 3–4 min, 30% A → 97% A; 4–6 min, 97% A. The flow rate was 0.3 mL/min, and the injection volume was set at 5 μL (*Ranjana et al., 2024*).

**Table 1 Escaphic acid and tiliroside spectrometry conditions.**

| NO | forluma | Name | m/z | Mass fragment | Cone(V) | Collision(V) | Precursor ion |
|---|---|---|---|---|---|---|---|
| 1 | $C_{19}H_{14}O$ | Escaphic acid | 488 | 487 | 38 | 10 | [M-H]- |
| | | | | 425 | 38 | 30 | |
| | | | | 407 | 38 | 42 | |
| 2 | $C_{30}H_{26}O_{13}$ | Tiliroside | 593 | 255 | 78 | 50 | [M-H]- |
| | | | | 284 | 38 | 42 | |
| | | | | 288 | 38 | 26 | |

*Mass spectrometry conditions*

The UPLC was coupled to a triple quadrupole mass spectrometer (TQ-MS) using electrospray ionization (ESI) in negative ion mode. The ion source temperature was set to 150 °C, with nitrogen as the cone gas at a flow rate of 50 L/h. Desolvation was achieved with nitrogen at 350 °C and a flow rate of 650 L/h. Data were collected using Multiple Reaction Monitoring (MRM) to quantify euscaphic acid and tiliroside, ensuring specificity and sensitivity. The target ion transitions and other mass spectrometry parameters for each analyte are listed below (Table 1) (*Li et al., 2014*; *Pieczykolan et al., 2019*; *Zan et al., 2018*).

Methodological investigation: euscaphic acid and tiliroside reserve solutions were diluted with methanol to prepare a series of concentration standard solutions. Measurements were carried out under the specified conditions to plot the standard curve. The determination method for total polyphenols was carried out according to the Folin-Ciocalteu method (*Lee & Lee, 2023*). To evaluate the precision, repeatability, stability, and sample recovery rate of the method, we conducted several experiments. Precision and repeatability were assessed by calculating the relative standard deviation (RSD) from six replicate injections and six independent samples, respectively. Stability testing was performed at 0 to 24 h after sample preparation, with RSD calculated. Sample recovery was determined by adding a known amount of standard to the matrix, followed by extraction and quantification, and calculating the ratio of the actual to theoretical concentration (*Kunc et al., 2023*).

## Establishment of a response surface model

### Single-factor experiment

Approximately 2.0000 g of FRL powder was weighed, and a single-factor experiment was conducted to optimize the extraction conditions. The effects of solvent concentration, extraction time, and solid-to-liquid ratio on extraction yield, total polyphenol content, and euscaphic acid content were investigated. In each experiment, only one factor was altered while the others remained constant. The extraction process was performed using either reflux or ultrasonic extraction, with specific conditions detailed in Table 2. The optimal extraction parameters were determined by evaluating the effects of each factor on extraction efficiency (*Heravi et al., 2022*). Based on the importance of the components, the evaluation is conducted using the following composite score formula: Composite

**Table 2 Design of single factor test.**

| Ethanol concentration (%) | Time (min) | Material-liquid ratio (g/mL) |
|---|---|---|
| 40 | 60 | 1:15 |
| 50 | 90 | 1:20 |
| 60 | 120 | 1:25 |
| 70 | 150 | 1:30 |
| 80 | 180 | 1:40 |

score = (Total polyphenol content/Maximum total polyphenol content) × 0.4 × 100 + (Euscaphic acid content/Maximum euscaphic acid content) × 0.4 × 100 + (Extraction yield/Maximum extraction yield) × 0.2 × 100. The Compositensive Score used in this study is a metric designed to comprehensively evaluate the effects of various extraction conditions. The calculation of the Compositensive Score takes into account the extraction efficiency of multiple target components (such as total polyphenols and euscaphic acid), with the aim of balancing the optimal extraction conditions for different components to derive an overall optimal extraction strategy. During the optimization process, the Compositensive Score helps ensure that the extraction rates of different target components reach a high level without overly prioritizing any single component.

### Response surface experiment

Based on the results of the single-factor experiment, the range of each factor in the response surface experiment was determined (Table 3). About 2.0000 g of FRL powder was weighed and subjected to reflux extraction. Data analysis was performed using Design Expert 11.0.1 (Stat-Ease, Inc., Minneapolis, MN, USA) to optimize the extraction process of FRL. The software employs RSM to evaluate the effects of multiple factors and their interactions on the extraction yield, total polyphenol content, and euscaphic acid content. Design Expert includes a built-in optimization function that uses desirability functions to predict the optimum parameters by maximizing or minimizing the response variables, depending on the study's objectives. This function allows for an accurate determination of the optimal extraction conditions. The optimized extraction process needs to be verified three times to confirm the accuracy of the optimization scheme (*Huang et al., 2023*; *Niu et al., 2021*).

### Establishment of the cell model

The evaluation of the Caco-2 cell model primarily involves using fluorescein sodium as a marker. The cells are seeded in a 12-well cell plate ($10^5$ cells/well) and observed and photographed every two days under a fluorescence inverted microscope. The transepithelial electrical resistance (TEER) is measured using a resistance meter (Millipore Millicell ERS-2; Millipore Sigma, Darmstadt, Germany). A TEER value of >400 $\Omega \cdot cm^2$ indicates that Caco-2 cells have formed a tight junction monolayer, suitable for transmembrane transport experiments (*Srinivasan et al., 2015*). Markers used for detection mainly include mannitol, inulin, and fluorescein sodium. When the apparent permeability

**Table 3 Experimental factor level and coding.**

| Level | Factor | | |
|---|---|---|---|
| | A Ethanol concentration/% | B Time/min | C Material-liquid ratio/mL/mg |
| −1 | 60 | 120 | 1:20 |
| 0 | 70 | 150 | 1:30 |
| 1 | 80 | 180 | 1:40 |

coefficient (*Papp*) of the marker is $<10^{-6}$ cm/s, it indicates that the integrity of the monolayer cells is good (*Li et al., 2021*).

## MTT assay

The MTT colorimetric method is used to investigate cell toxicity. The detection principle is that succinate dehydrogenase in the mitochondria can reduce MTT to water-insoluble blue-violet formazan crystals, while dead cells lack this function. Caco-2 cells at a density of $1 \times 10^4$ cells/mL in the logarithmic growth phase are inoculated into each well of a 96-well plate (200 μL per well) and cultured for 24 h. After removing the culture medium, a complete culture medium with different concentrations of FRLE is added to each well, with six replicates for each concentration. The control group is given a complete culture medium without the drug. After incubation for 24, 48, and 72 h, 20 μL of 5 mg/mL MTT solution is added to each well and cultured for another 4 h. The culture medium is then discarded, and 150 μL of DMSO is added to dissolve the blue-violet formazan crystals in each well. After thorough mixing, the absorbance is measured at 570 nm using a microplate reader, and the cell survival rate is calculated (*Alizadeh-Navaei et al., 2016*).

## Absorption characteristics study

A precise amount of FRLE reserve solution is diluted to concentrations of 50, 100, and 150 μg/mL, representing the low, medium, and high concentrations of FRLE. Additionally, solutions are prepared containing 0.100 μmoL/mL Verapamil (Ver), 0.099 μmoL/mL Cyclosporine A (CsA), and 0.025 μmoL/mL EGTA at low, medium, and high concentrations for bidirectional transport.

Caco-2 cells are seeded at a density of $2.5 \times 10^5$ cells/mL, with 0.5 mL inoculated into each well of a Transwell® 12-well plate. After continuous culturing for 18 days, the cell model is established. Bidirectional transport experiments include AP→BL (B→A, absorption transport) and BL→AP (A→B, secretion transport).

Taking absorption transport as an example: 0.5 mL of drug-containing Hank's Balanced Salt Solution (HBSS) solution is added to the apical (AP) side (supply side), while 1.5 mL of blank solution is added to the basolateral (BL) side (receiving side). At time points 15, 30, 60, 90, 120, 150, and 180 min, 200 μL is drawn from the BL side and simultaneously supplemented with 200 μL of HBSS buffer. The samples are then evaporated in a water bath, re-dissolved in methanol, and centrifuged at 12,000 rpm for 10 min. The supernatant

is taken and assayed under the conditions described in "Establishment of the determination method for euscaphic acid and tiliroside". The effects of different times, concentrations, inhibitors, chelating agents, and pH values on the absorption characteristics of euscaphic acid and tiliroside in FRLE are investigated. The secretion transport procedure is identical to the absorption transport (*Artursson & Karlsson, 1991*).

## Calculation formulas and statistical analysis

The side with the added blank solution is considered the receiving side, and the amount of drug in the receiving side is regarded as its absorption amount ($Qr$). In the A→B transport test, the amount of drug absorbed by the B-side chamber at any time point is denoted as ($QrBi$). In the B→A transport test, the amount of drug absorbed by the A-side chamber at any time point is ($QrAi$). The specific calculations are provided in the following formulas:

A-B transport: $Q_{rBi} = 0.2 \times (C_{r1} + C_{r2} + \Lambda + C_{r(i-1)}) + 1.5 \times C_{ri}$

B-A transport: $Q_{rdi} = 0.2 \times (C_{r1} + C_{r2} + \Lambda + C_{r(i-1)}) + 0.5 \times C_{di}$

In the formula, 1.5 represents the volume of the test solution added to the B-side chamber (mL), 0.5 represents the volume of the test solution added to the A-side chamber (mL), and 0.2 is the volume of each sample taken (mL). Cri is the actual concentration in the receiving chamber at the i-th time point (µg/mL). The apparent permeability coefficient *Papp* (cm/s) is calculated using the following formula (*Lu et al., 2020*):

$$Papp = \frac{dQ}{dt} \times \frac{1}{A \times C_0}$$

dQ/dt represents the amount of drug transported per unit of time (µg/s); A is the surface area of the 12-well Transwell® membrane (1.12 cm$^2$), and C$_0$ is the initial concentration of the drug in the donor side (µg/cm) (*Qu et al., 2021*).

Exhaust ratio (R$_{B-A/A-B}$)) is calculated using the following formula:

$$R_{B-A/A-B} = \frac{Papp(B-A)}{Papp(A-B)}$$

SPSS 20.1 software (IBM, Armonk, NY, USA) was used for analysis of variance (ANOVA) on the data. Results are represented by $\bar{x} \pm s$. The t-test was employed for intergroup difference comparison. When $P < 0.05$, the results are considered statistically significant.

# RESULTS

## Methodology establishment for euscaphic acid, tiliroside, and total polyphenols

The standard curve for euscaphic acid is A = 30,214X + 19,246 (r = 0.9999; 0.1~32 µg/mL; LOD = 0.03 µg/mL, LOQ = 0.10 µg/mL). The standard curve for tiliroside is A = 573833x − 3040.9 (r = 0.999; 0.005~1 µg/mL; LOD = 0.0015 µg/mL, LOQ = 0.005 µg/mL). The standard curve for gallic acid is A = 108.03x + 0.1896 (r = 0.999; 10.30 ~ 92.70 µg/mL; LOD = 3.09 µg/mL, LOQ = 10.30 µg/mL) (Figs. 1, 2).

The precision, repeatability, and stability of euscaphic acid, tiliroside, and gallic acid were evaluated, with all RSD values being <2.00%, indicating good consistency. Recovery

## Euscaphic acid

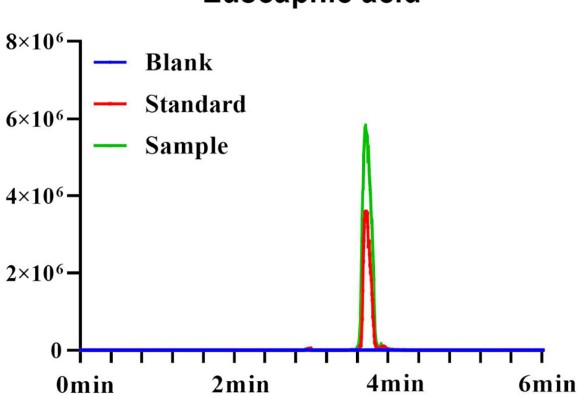

## Tiliroside

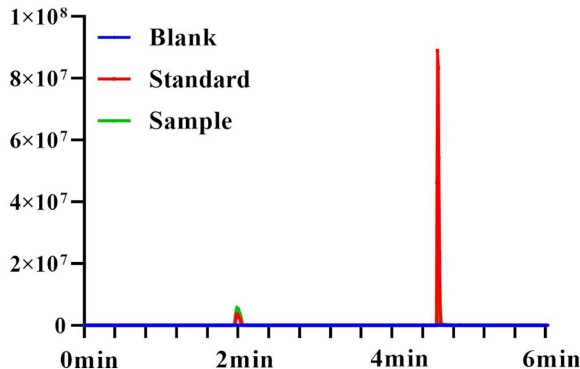

**Figure 1** **UPLC plots of euscaphic acid and tiliroside.**

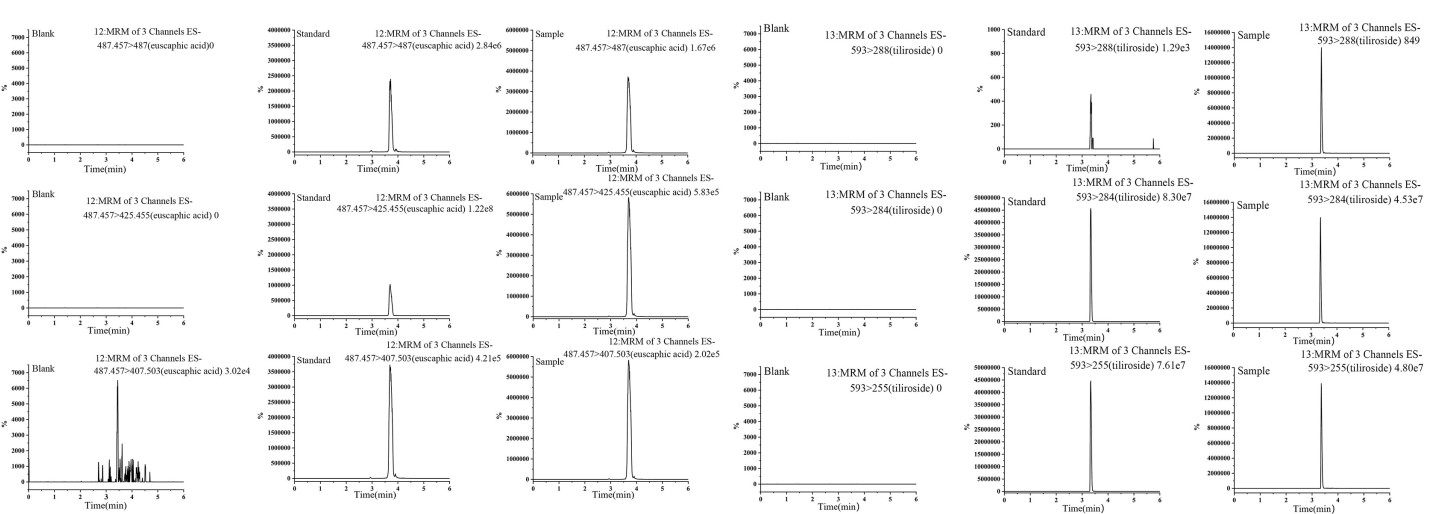

**Figure 2** **MRM plots of euscaphic acid and tiliroside.**

**Table 4 Results of sample adding recovery ($n = 9$, $\bar{x} \pm s$).**

| Component | Sample content (mg) | Add content (mg) | Detected levels (mg) | Recovery rate (%) | Average recovery rate (%) | RSD (%) |
|---|---|---|---|---|---|---|
| Euscaphic acid | 1.21 | 1.00 | 2.19 | 99.12 | 99.75 ± 1.14 | 1.14 |
| | 1.22 | 1.00 | 2.18 | 98.12 | | |
| | 1.15 | 1.00 | 2.16 | 100.62 | | |
| | 1.17 | 1.00 | 2.20 | 100.95 | | |
| | 1.17 | 1.00 | 2.15 | 99.08 | | |
| | 1.20 | 1.00 | 2.22 | 100.63 | | |
| Tiliroside | 1.22 | 1.20 | 2.54 | 104.76 | 100.22 ± 2.79 | 2.78 |
| | 1.21 | 1.20 | 2.44 | 101.36 | | |
| | 1.26 | 1.20 | 2.39 | 97.03 | | |
| | 1.28 | 1.20 | 2.51 | 101.16 | | |
| | 1.22 | 1.20 | 2.38 | 98.35 | | |
| | 1.25 | 1.20 | 2.42 | 98.67 | | |
| Gallic acid | 49,830 | 50,000 | 98,400 | 98.57 | 99.08 ± 0.56 | 0.57 |
| | 49,010 | 50,000 | 97,970 | 98.95 | | |
| | 48,980 | 50,000 | 98,050 | 99.06 | | |
| | 49,870 | 50,000 | 98,480 | 98.61 | | |
| | 49,970 | 50,000 | 100,080 | 100.11 | | |
| | 48,570 | 50,000 | 97,780 | 99.20 | | |

studies were conducted by spiking known amounts of each standard into the sample matrix. The average recovery rate for euscaphic acid was (99.75 ± 1.14)% with an RSD of 1.14%, for tiliroside it was (100.22 ± 2.78)% with an RSD of 2.78%, and for gallic acid it was (99.08 ± 0.56)% with an RSD of 0.57%. These results demonstrate the high accuracy of the analytical method, confirming both the efficiency of the extraction and the reliability of the analysis (Table 4).

## Response surface experiment

The first step in the response surface experiment is a single-factor investigation. This mainly includes solid-liquid ratio, ethanol concentration, and extraction time. When the solid-liquid ratio increased from 1:15 (g/mL) to 1:30 (g/mL), the composite score significantly increased, rising from 75.53 to 92.38. Further increasing the solid-liquid ratio resulted in a significant decrease in the composite score. Using 60% ethanol as the extraction solvent yielded the highest composite score of 91.49. As the reflux extraction time extended, the composite score first increased and then decreased, with the highest score of 93.34 achieved at 150 min (Fig. 3A–3C).

Subsequently, the Box-Behnken design using Design Expert 11 software was used to design the experimental scale (Table 5) and calculate the composite scores. Based on the established model, 3D response surface plots and contour plots illustrating the effects of various influencing factors on the composite score were generated (Fig. 3D–3I). The figures show that the interaction of feed/liquid ratio and extraction time had a significant

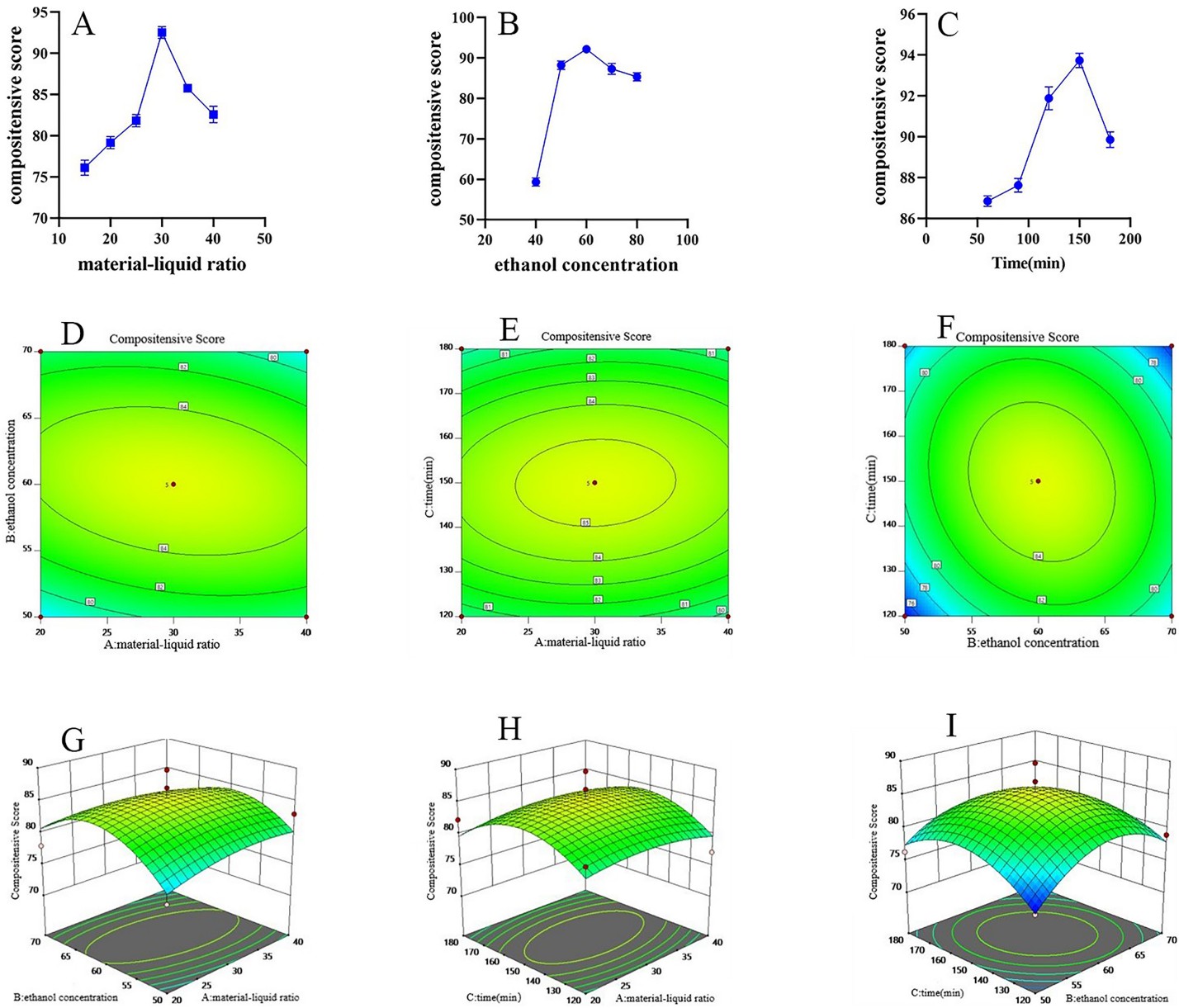

**Figure 3 FRL response surface methodology research.** (A) Effect of material liquid ratio on extraction rate ($n = 3$, $\bar{x} \pm s$). (B) Effect of ethanol concentration on extraction rate ($n = 3$, $\bar{x} \pm s$). (C) Effect of reflux time on extraction rate ($n = 3$, $\bar{x} \pm s$). (D–F) FRL response surface contour plot. (G–I) FRL response surface contour 3D plot.

impact on extraction efficiency, followed by the interaction of ethanol concentration and extraction time. While drawing the contour plots, a binomial regression model was also obtained. The formula for the composite score is:

$$\text{Composite Score} = 85.45 + 0.0122 \times A + 0.2461 \times B - 0.0002 \times C\text{-}1.13 \times AB \\ + 0.2982 \times AC\text{-}1.29 \times BC\text{-}1.24 \times A2\text{-}4.89 \times B2\text{-}4.24 \times C2$$

Based on the model, it can be deduced that A, B, C, A2, AB, AC, and BC are not significant; B2 and C2 are significant (Table 6). Based on the above results, the optimal

**Table 5 Experimental design and results of response surface method.**

| No. | A | B | C | Factor | | | |
|-----|---|---|---|--------|--------|--------|--------|
| | | | | Extract volume (g) | Total polyphenols (mg/g) | Euscaphic acid (mg/g) | Compositensive score |
| 1 | −1 | −1 | 0 | 0.80 | 171.22 | 2.55 | 81.90 |
| 2 | −1 | 0 | 1 | 0.73 | 185.48 | 2.13 | 77.95 |
| 3 | 0 | 0 | 0 | 0.90 | 144.18 | 2.80 | 81.94 |
| 4 | 0 | 1 | 1 | 0.83 | 174.48 | 1.94 | 75.44 |
| 5 | 0 | 0 | 0 | 0.89 | 154.80 | 2.81 | 83.90 |
| 6 | 0 | 0 | 0 | 0.90 | 190.17 | 2.32 | 84.78 |
| 7 | −1 | 1 | 0 | 0.83 | 207.34 | 1.98 | 82.27 |
| 8 | 1 | −1 | 0 | 0.94 | 189.30 | 1.67 | 77.11 |
| 9 | 1 | 0 | −1 | 0.98 | 202.52 | 1.86 | 82.95 |
| 10 | 1 | 0 | 1 | 0.97 | 158.33 | 2.32 | 79.94 |
| 11 | 0 | 0 | 0 | 0.87 | 166.17 | 3.12 | 89.66 |
| 12 | −1 | 0 | −1 | 0.81 | 203.92 | 1.61 | 76.45 |
| 13 | 0 | 1 | −1 | 0.98 | 142.95 | 2.25 | 76.29 |
| 14 | 0 | −1 | 1 | 0.85 | 156.79 | 2.45 | 78.93 |
| 15 | 0 | 0 | 0 | 0.90 | 178.31 | 2.53 | 84.96 |
| 16 | 1 | 1 | 0 | 0.90 | 170.26 | 2.14 | 78.62 |
| 17 | 0 | −1 | −1 | 0.93 | 184.98 | 1.56 | 74.61 |

**Table 6 Variance analysis of response surface regression model.**

| Source | Sum of squares | df | Mean square | F-value | p-value |
|--------|----------------|----|-----------| --------|---------|
| **Model** | 211.61 | 9 | 23.51 | 2.18 | 0.1585 |
| A- material-liquid ratio | 0.0012 | 1 | 0.0012 | 0.0001 | 0.9919 |
| B- ethanol concentration | 0.4847 | 1 | 0.4847 | 0.0449 | 0.8382 |
| C-Time | $2.450 \times 10^{-7}$ | 1 | $2.450 \times 10^{-7}$ | $2.271 \times 10^{-8}$ | 0.9999 |
| AB | 5.09 | 1 | 5.09 | 0.4720 | 0.5142 |
| AC | 0.3557 | 1 | 0.3557 | 0.0330 | 0.8611 |
| BC | 6.67 | 1 | 6.67 | 0.6181 | 0.4575 |
| $A^2$ | 6.50 | 1 | 6.50 | 0.6020 | 0.4632 |
| $B^2$ | 100.53 | 1 | 100.53 | 9.32 | 0.0185 |
| $C^2$ | 75.81 | 1 | 75.81 | 7.03 | 0.0329 |
| **Residual** | 75.53 | 7 | 10.79 | | |
| Lost proposal | 40.36 | 3 | 13.45 | 1.53 | 0.3366 |
| Pure error | 35.17 | 4 | 8.79 | | |
| **Total** | 287.14 | 16 | | | |

extraction conditions were determined to maximize extraction yield, total polyphenol content, and Euscaphic acid content. The initial conditions were identified as a solid-liquid ratio of 1:35.72 g/mL, 67.14% ethanol, and extraction time of 135.34 min. For practical

convenience, the parameters were adjusted to a solid-liquid ratio of 1:35 g/mL, 65% ethanol, and a reflux time of 135 min.

The optimized extraction process was verified by conducting three repetitions. The results showed that the average extraction yield was 51.00 ± 1.07%, the average total polyphenol content was 126.55 ± 2.61 mg/g, and the average euscaphic acid content was 2.90 ± 0.08 mg/g. The composite score, which serves as a comprehensive measure of extraction efficiency for all target components, was 82.38, with an average deviation of 1.01% from the predicted value of 80.217. This indicates that the extraction model meets the experimental criteria and is suitable for achieving the desired outcomes across multiple response variables.

## Evaluation of the Caco-2 cell model and MTT assay

The cultured Caco-2 cells were observed and photographed under a fluorescent inverted microscope on days 5, 9, 13, and 18 (Fig. 4A). The results show that by the 18th day, the cells were closely aligned, indicating the successful establishment of the Caco-2 cell monolayer model. TEER value results demonstrate a continuous rise with increased culture time, stabilizing around the 20th day. With a TEER value >400 $\Omega/cm^2$, it suggests that the electrical resistance of the Caco-2 cell monolayer model meets the experimental requirements (Fig. 4B). Concurrently, the $Papp$ of fluorescein sodium was determined through four repeated measurements, showing a $Papp$ of $(4.06 ± 0.17) \times 10^{-7}$ cm/s. The results of these three validations indicate that the Caco-2 cell model was successfully established. The cell survival rate was calculated as follows: (OD value of experimental wells—OD value of blank wells)/(OD value of control wells-OD value of blank wells) × 100%. The safe concentration range of FRLE is 0.5 ~ 100 µg/mL (Fig. 4C).

## Absorption characteristics experiment of euscaphic acid

The $Qr$ of euscaphic acid in FRLE is proportional to its concentration. There is a significant difference between the $Qr$ from A to B (A→B) and the $Qr$ from B to A (B→A). The $Papp$ of euscaphic acid is greater than $10^{-6}$, indicating that euscaphic acid is well absorbed. The $Papp$ value of transport from A→B is significantly different from the $Papp$ value of transport from B→A. There is no significant difference in the absorption amount on the AP side compared to the BL side, indicating that the transport mechanism of euscaphic acid is passive (Table 7, Figs. 5A–5C). After the addition of the P-glycoprotein (P-gp) inhibitor, the $Qr$ and $Papp$ of euscaphic acid in FRLE significantly increased, with an $R_{B→A/A→B}$ value around 1.20. This suggests that some components in FRLE combine with euscaphic acid to form substrates for P-gp, resulting in an increased $Qr$ of euscaphic acid when the P-gp inhibitor is added. Moreover, the results also show that there isn't a significant difference in bidirectional transport (Table 7, Figs. 5D–5F). After the addition of the EGTA chelating agent, the transport amount and $Papp$ value of euscaphic acid in FRLE showed a significant increase, with an $R_{B→A/A→B}$ value ranging between 0.8 and 1.4. This indicates that some component in FRLE interacts with euscaphic acid, serving as a substrate for the EGTA chelating agent, facilitating paracellular transport (Table 7, Figs. 5G–5I). In an acidic environment (pH 6.0), the $Qr$ and $Papp$ values of euscaphic acid in FRLE are the highest,

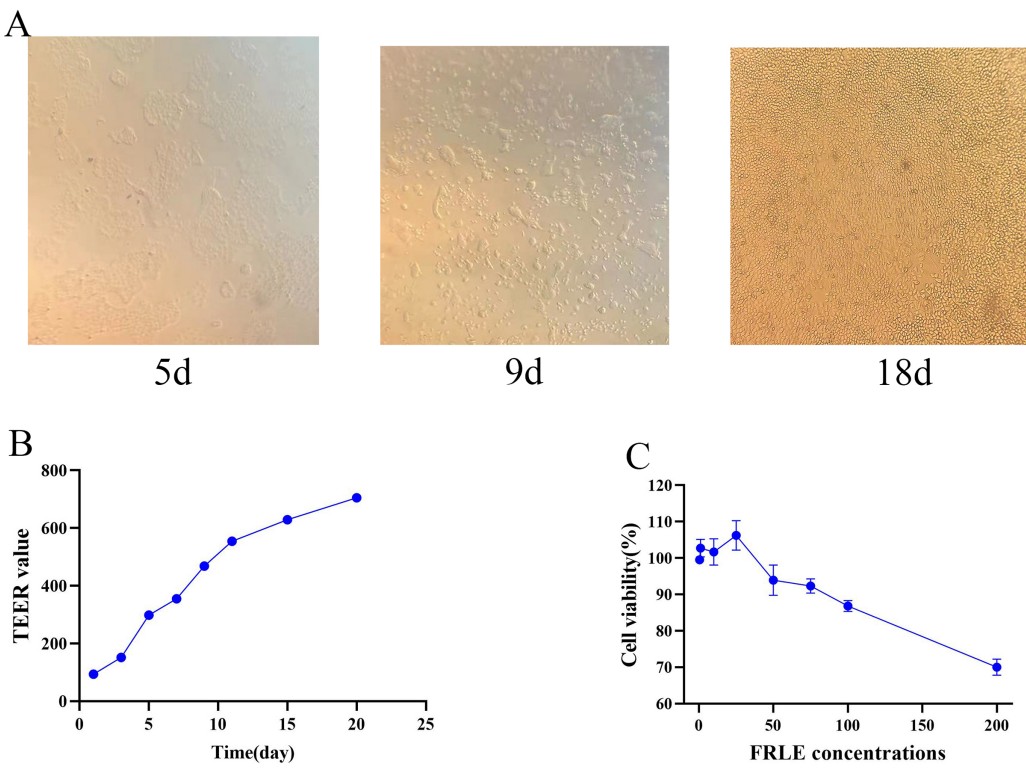

**Figure 4 Establishment of cell models.** (A) Pictures of Caco-2 cells at different time points. (B) TEER values at different time points ($n = 3$). (C) Survival of Caco-2 cells at different concentrations of FRLE ($n = 3$).

**Table 7 Bidirectional transport of euscaphic acid in FRLE in Caco-2 cells by *Papp* and efflux ratio (ER) (P-gp inhibitors; EGTA; different pH values) ($n = 3$, x̄ ± s).**

| Euscaphic acid | C (μg/mL) | *Papp* × 10⁻⁴ (cm/s) (A→B) | *Papp* × 10⁻⁴ (cm/s) (B→A) | $R_{B→A/A→B}$ |
|---|---|---|---|---|
| FRLE | 50.00 | 41.09 ± 0.26 | 36.17 ± 0.34** | 0.88 ± 0.01 |
| | 100.00 | 45.62 ± 0.28 | 39.10 ± 0.27** | 0.86 ± 0.01 |
| | 150.00 | 56.94 ± 7.10 | 77.50 ± 2.90** | 1.38 ± 0.21 |
| FRLE+Ver | 50.00 | 122.03 ± 2.77** | 87.85 ± 4.64** | 0.72 ± 0.01 |
| | 100.00 | 113.97 ± 0.94** | 88.65 ± 6.21** | 0.78 ± 0.06 |
| | 150.00 | 113.56 ± 1.84** | 112.74 ± 2.67** | 0.99 ± 0.01 |
| FRLE+CsA | 50.00 | 67.34 ± 3.74** | 83.88 ± 5.12** | 1.25 ± 0.03 |
| | 100.00 | 89.58 ± 4.38** | 89.97 ± 0.70** | 1.00 ± 0.04 |
| | 150.00 | 104.34 ± 0.41** | 164.41 ± 1.32** | 1.58 ± 0.01 |
| FRLE+EGTA | 50.00 | 89.98 ± 3.79** | 90.59 ± 4.60** | 1.01 ± 0.05 |
| | 100.00 | 105.04 ± 1.39** | 101.59 ± 1.81** | 0.97 ± 0.02 |
| | 150.00 | 167.54 ± 10.09** | 204.66 ± 0.67** | 1.22 ± 0.07 |
| FRLE+pH5.0 | 50.00 | 62.79 ± 0.43** | 72.63 ± 5.71** | 1.16 ± 0.08 |
| | 100.00 | 153.07 ± 6.91** | 177.52 ± 6.70** | 1.16 ± 0.03 |
| | 150.00 | 203.43 ± 7.70** | 196.43 ± 6.72** | 0.97 ± 0.06 |

(Continued)

| Euscaphic acid | C (µg/mL) | $Papp \times 10^{-4}$ (cm/s) (A→B) | $Papp \times 10^{-4}$ (cm/s) (B→A) | $R_{B \to A/A \to B}$ |
|---|---|---|---|---|
| **Table 7 (continued)** | | | | |
| FRLE+pH6.0 | 50.00 | 149.71 ± 9.55** | 141.79 ± 5.11** | 0.95 ± 0.05 |
| | 100.00 | 220.69 ± 6.34** | 262.09 ± 4.31** | 1.19 ± 0.02 |
| | 150.00 | 262.87 ± 23.29** | 268.75 ± 3.56** | 1.03 ± 0.09 |
| FRLE+pH8.0 | 50.00 | 23.11 ± 1.15** | 20.57 ± 3.15** | 1.12 ± 0.03 |
| | 100.00 | 28.69 ± 2.14** | 30.14 ± 2.76** | 0.95 ± 0.02 |
| | 150.00 | 33.12 ± 3.21** | 36.24 ± 1.03** | 0.91 ± 0.05 |
| FRLE+pH9.0 | 50.00 | 8.21 ± 0.89** | 7.54 ± 1.11* | 1.08 ± 0.03 |
| | 100.00 | 9.32 ± 1.01** | 8.02 ± 0.57** | 1.16 ± 0.04 |
| | 150.00 | 9.87 ± 0.69** | 8.65 ± 0.87** | 1.14 ± 0.01 |

Notes:
Compared with $Qr$(µg) (A→B) or $Papp \times 10^{-4}$ (cm/ s) (A→B).
* $P < 0.05$.
** $P < 0.01$.

with an $R_{B \to A/A \to B}$ value around 1.0. The absorption is lowest in a neutral solution (pH 7.2), indicating that an acidic environment is favorable for the transport of euscaphic acid (Table 7, Figs. 5J–5O).

## Absorption characteristics experiment of tiliroside

The $Qr$ and $Papp$ of tiliroside in FRLE are proportional to its concentration, and the $Papp$ is greater than $10^{-6}$, indicating that tiliroside is well absorbed. The transport from A to B (A-B) is significantly lower than the $Papp$ value of transport from B to A (B-A), suggesting that tiliroside's transport mechanism is passive (Table 8, Figs. 6A–6C). Upon adding the P-gp inhibitor, the bidirectional transport $Qr$ of tiliroside in FRLE significantly increased, with an $R_{B \to A/A \to B}$ value ranging from 0.7 to 1.3. This indicates that tiliroside in FRLE is a substrate for P-gp, and there's no significant difference in its bidirectional transport (Table 8, Figs. 6D–6F).

After the addition of the EGTA chelating agent, the $Qr$ and $Papp$ of tiliroside in FRLE significantly increased. The $R_{B \to A/A \to B}$ value of tiliroside in FRLE ranged between 0.9 and 1.2, suggesting no noticeable difference in bidirectional transport (Table 8, Figs. 6G–6I). In an acidic environment (pH 6.0), the $Qr$ and $Papp$ of tiliroside in FRLE were the highest, while the absorption was lowest in a neutral solution (pH 7.2). The $R_{B \to A/A \to B}$ value of tiliroside in FRLE was around 1.0, indicating that an acidic environment is favorable for the transport of tiliroside, with no noticeable directionality in its transport (Table 8, Figs. 6J–6O).

## DISCUSSION

The choice of ultra-high performance liquid chromatography-triple quadrupole mass spectrometry (UPLC-TQ-MS) was based on its high sensitivity and accuracy, allowing for precise quantification of euscaphic acid and tiliroside in complex samples. The experimental results demonstrated that the standard curves for euscaphic acid and tiliroside showed excellent linearity (r = 0.9999 and r = 0.999), and the limits of detection

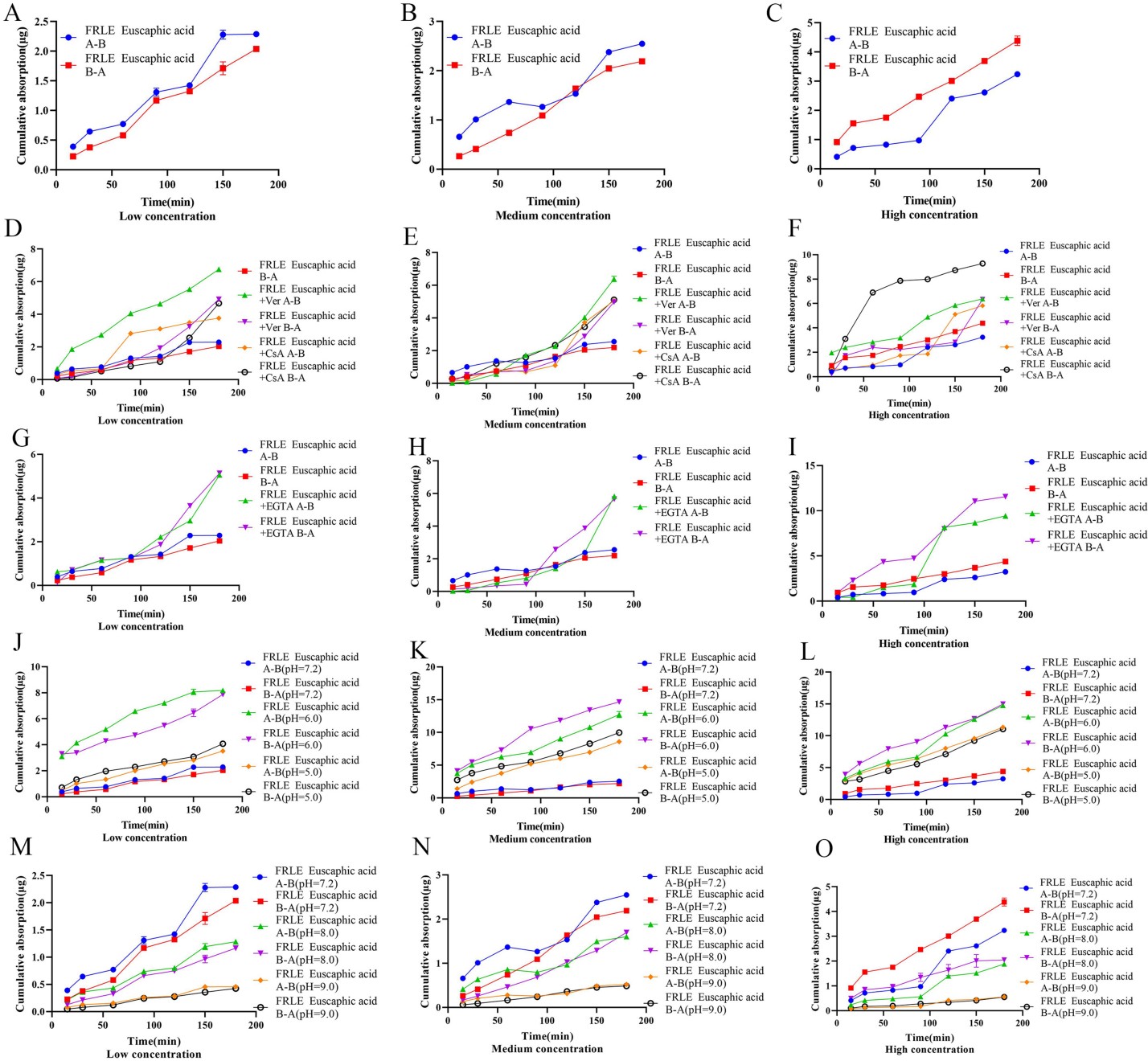

**Figure 5 Cumulative uptake of euscaphic acid in different environments.** (A–C) Schematic representation of cumulative transporter uptake of euscaphic acid in FRLE (low, medium, and high concentrations); (D–F) effect of inhibitors on the cumulative transporter uptake of euscaphic acid in FRLE (low, medium, and high concentrations); (G–I) effect of EGTA on the cumulative transporter uptake of euscaphic acid in FRLE (low, medium, and high concentrations); (J–O) effects of different pH environments on the cumulative translocation uptake of euscaphic acid in FRLE (low, medium, and high concentrations).

and quantification further confirmed the method's capability in detecting the samples. The optimization of extraction methods aimed to enhance the extraction efficiency of the target compounds, ensuring the efficiency and reliability of the experiment. Through single-factor experiments, we identified the optimal conditions of 70% solvent

**Table 8 Bidirectional transport of tiliroside in FRLE in Caco-2 cells by *Papp* and efflux ratio (ER) (P-gp inhibitors; EGTA; Different pH values) ($n = 3$, $\bar{x} \pm s$).**

| Tiliroside | C (μg/mL) | *Papp* × 10$^{-4}$ (cm/s) (A→B) | *Papp* × 10$^{-4}$ (cm/s) (B→A) | R$_{B→A/A→B}$ |
|---|---|---|---|---|
| FRLE | 50.00 | 1.40 ± 0.02 | 1.15 ± 0.04** | 0.82 ± 0.04 |
| | 100.00 | 1.84 ± 0.11 | 1.39 ± 0.08** | 0.75 ± 0.07 |
| | 150.00 | 2.06 ± 0.08 | 2.31 ± 0.04* | 1.12 ± 0.03 |
| FRLE+Ver | 50.00 | 1.42 ± 0.02* | 1.56 ± 0.03** | 1.09 ± 0.03 |
| | 100.00 | 1.88 ± 0.08# | 2.32 ± 0.06** | 1.24 ± 0.08 |
| | 150.00 | 2.51 ± 0.06** | 2.40 ± 0.02* | 0.96 ± 0.02 |
| FRLE+CsA | 50.00 | 1.46 ± 0.01* | 1.55 ± 0.02** | 1.06 ± 0.03 |
| | 100.00 | 1.80 ± 0.05# | 2.26 ± 0.09** | 1.25 ± 0.06 |
| | 150.00 | 2.61 ± 0.06** | 2.76 ± 0.01** | 1.03 ± 0.01 |
| FRLE+EGTA | 50.00 | 1.37 ± 0.02# | 1.29 ± 0.02* | 0.95 ± 0.01 |
| | 100.00 | 1.76 ± 0.13# | 2.10 ± 0.04** | 1.20 ± 0.12 |
| | 150.00 | 2.68 ± 0.09** | 2.83 ± 0.03** | 1.06 ± 0.03 |
| FRLE+pH5.0 | 50.00 | 3.80 ± 0.09** | 4.48 ± 0.08** | 1.18 ± 0.01 |
| | 100.00 | 4.09 ± 0.07** | 5.36 ± 0.14** | 1.31 ± 0.02 |
| | 150.00 | 6.49 ± 0.08** | 7.18 ± 0.24** | 1.11 ± 0.05 |
| FRLE+pH6.0 | 50.00 | 4.73 ± 0.24** | 5.68 ± 0.24** | 1.20 ± 0.11 |
| | 100.00 | 4.68 ± 0.14** | 6.14 ± 0.08** | 1.31 ± 0.05 |
| | 150.00 | 6.18 ± 0.28** | 8.08 ± 0.15** | 1.21 ± 0.15 |
| FRLE+pH8.0 | 50.00 | 0.86 ± 0.12** | 0.83 ± 06** | 1.03 ± 0.01 |
| | 100.00 | 0.92 ± 0.17** | 0.88 ± 0.11** | 1.04 ± 0.03 |
| | 150.00 | 0.96 ± 0.14** | 0.93 ± 0.18** | 1.03 ± 0.05 |
| FRLE+pH9.0 | 50.00 | 0.45 ± 0.04** | 0.42 ± 0.15** | 1.07 ± 0.06 |
| | 100.00 | 0.56 ± 0.24** | 0.54 ± 0.18** | 1.03 ± 0.04 |
| | 150.00 | 0.66 ± 0.26** | 0.62 ± 0.21** | 1.06 ± 0.05 |

**Notes:**
Compared with $Qr$(μg) (A→B) or *Papp* × 10$^{-4}$ (cm/ s) (A→B).
# $P > 0.05$.
* $P < 0.05$.
** $P < 0.01$.

concentration, 90-min extraction time, and a solid-to-liquid ratio of 1:30. These conditions significantly improved the extraction efficiency and the content of the target compounds, thereby ensuring the accuracy and reproducibility of downstream analyses. In our previous studies, we found that the total polyphenol content in rosehip was significantly higher than other components, which led to the selection of total polyphenols as a key indicator for evaluating extraction efficiency. The choice of total polyphenols as the main criterion was due to their ability to reflect the comprehensive antioxidant activity of rosehip, aligning with our goal of optimizing extraction methods to maximize the yield of bioactive compounds. Regarding the selection of individual components, mass spectrometry analysis revealed that euscaphic acid was present in relatively high concentrations (*Wang, Xie & Tian, 2024*). Research on this compound in the current literature is limited, highlighting its potential for further exploration. Euscaphic acid plays an important role in

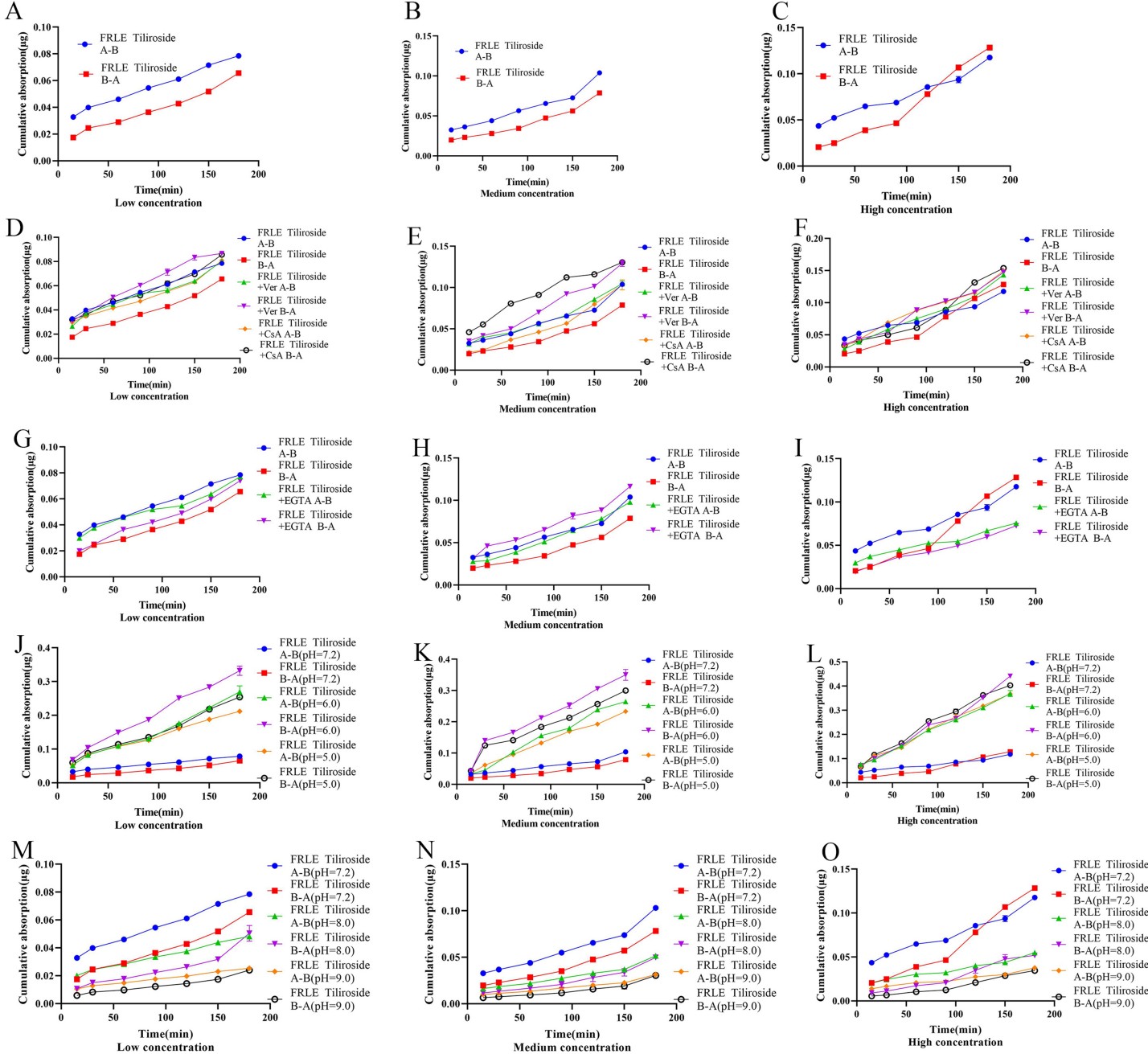

**Figure 6 Cumulative uptake of tiliroside in different environments.** (A–C) Schematic representation of cumulative transporter uptake of tiliroside in FRLE (low, medium, and high concentrations); (D–F) effect of inhibitors on the cumulative transporter uptake of tiliroside in FRLE (low, medium, and high concentrations); (G–I) effect of EGTA on the cumulative transporter uptake of tiliroside in FRLE (low, medium, and high concentrations); (J–O) effects of different pH environments on the cumulative translocation uptake of tiliroside in FRLE (low, medium, and high concentrations).

our study, and the lack of research on its biological activity supports the rationale for deeper investigation into this compound. We believe that euscaphic acid warrants further study to explore its bioactivity and potential therapeutic applications in the future.

The Caco-2 cell model is widely used to study drug absorption and transport. Upon reaching peak growth, these cells differentiate to form distinct AP and BL sides (*Chantret et al., 1988*). The AP side, representing the luminal side of the intestine, contains intestinal microvillus hydrolases and carriers, whereas the BL side corresponds to the inner wall of the intestine (*Hilgers, Conradi & Burton, 1990*). A key distinction between these sides is the higher level of alkaline phosphatase on the AP side compared to the BL side (*Panse & Gerk, 2022*). In this study, we established a Caco-2 monolayer absorption model and assessed various indicators, including cell morphology, TEER, and fluorescein sodium permeability (*Papp*). The results confirmed the successful establishment of the model, meeting the requirements for subsequent uptake and transport experiments (*Shah et al., 2006*).

The Caco-2 monolayer cell absorption model was used to study the absorption characteristics of the drug, with cell viability required to be greater than 85%. When cell viability falls below 85%, the TEER of the Caco-2 monolayer significantly decreases (*Kamiloglu et al., 2015*). The MTT assay was used to assess the cytotoxicity of FRLE on Caco-2 cells. Results indicated that at FRLE concentrations of 150 μg/mL or lower, cell viability remained above 85%. Consequently, the low, medium, and high concentrations for subsequent experiments were set at 50, 100, and 150 μg/mL, respectively, meeting the experimental requirements.

The *Papp* reflects a substance's absorptive capacity in the intestinal tract. Substances absorbed at rates of 0% to 100% in the human jejunum have *Papp* values ranging from $5 \times 10^{-8}$ to $5 \times 10^{-5}$ cm/s in Caco-2 cell transport (*Ozeki et al., 2015*). Drugs with good absorption have *Papp* values greater than $1 \times 10^{-5}$ cm/s; those with moderate absorption have *Papp* values between $1 \times 10^{-6}$ and $1 \times 10^{-5}$ cm/s; and drugs with poor absorption have *Papp* values less than $1 \times 10^{-6}$ cm/s (*Lee et al., 2017*; *Nakazono et al., 2021*). The results of this study show that in FRLE, the *Papp* values for euscaphic acid and tiliroside are positively correlated with concentration, ranging from $(7.54 \pm 1.11) \times 10^{-4}$ to $(268.75 \pm 3.56) \times 10^{-4}$ cm/s and $(0.42 \pm 0.15) \times 10^{-4}$ to $(8.08 \pm 0.15) \times 10^{-4}$ cm/s, respectively, categorizing them as substances with good absorption.

In the Caco-2 cell absorption model, the $R_{B \to A/A \to B}$ ratio can be used to evaluate the bidirectional transport directionality of a drug. An $R_{B \to A/A \to B}$ ratio close to 1 indicates that the drug is primarily transported by passive diffusion. When the $R_{B \to A/A \to B}$ ratio is much less than 1 (generally <0.6), it suggests that the drug is predominantly taken up by transporters on the apical side of the small intestine (*Jin et al., 2016*). Conversely, an $R_{B \to A/A \to B}$ ratio significantly greater than 1 (generally >1.5) indicates that the drug is actively effluxed by transport proteins on the apical side of the small intestine. In this study, the $R_{B \to A/A \to B}$ ratios for euscaphic acid and tiliroside in FRLE ranged from 0.72 to 1.58 and 0.75 to 1.31, respectively, suggesting that their transport is primarily governed by passive diffusion. P-gp is a membrane protein with a molecular weight of $1.7 \times 10^5$, encoded by the multidrug resistance (MDR) gene. It is located on the villous side of the cell and acts as an energy-dependent drug efflux pump (*Nagayasu et al., 2019*). P-gp is located on the apical side of Caco-2 cells, and when drugs are transported, the transport rates of A→B and B→A decrease. P-gp inhibitors can suppress the antagonistic effect of P-gp on

drug absorption, thereby promoting drug uptake (*Broccatelli, 2012*; *Ferreira et al., 2021*). In this study, two typical P-gp inhibitors, Ver and CsA, were selected to investigate their effects on the transport of euscaphic acid and tiliroside from FRLE in Caco-2 cells. The results showed that after adding P-gp inhibitors to FRLE, the *Papp* values of bidirectional transport of euscaphic acid and tiliroside both increased correspondingly. This indicates that euscaphic acid and tiliroside in FRLE are substrates of P-gp. Both euscaphic acid and tiliroside, as well as FRLE, have an efflux ratio (ER) <1.5, indicating no apparent directional transport.

The primary modes of connection between intestinal epithelial cells are adherens junctions, tight junctions, and desmosomes. The integrity of these connections largely depends on the presence of $Ca^{2+}$. EGTA is a specific calcium ion chelator that can open intercellular gaps, significantly enhancing drug transport capability (*Appel (Kohn) et al., 2021*; *O'Doherty et al., 2020*). Research findings indicate that after the addition of EGTA to the Caco-2 cell model, there was no significant change in the bidirectional transport *Papp* values for euscaphic acid in FRLE; however, there was a substantial increase in the *Papp* values for tiliroside, suggesting that the transport mechanism for tiliroside involves paracellular transport.

In this study, the effects of different pH levels on the absorption characteristics of euscaphic acid and tiliroside in FRLE were investigated. According to intestinal absorption theory, drugs, once ionized, are less likely to be absorbed through the intestinal barrier, primarily due to changes in solvent pH. The results showed that the highest *Qr* and *Papp* values for Euscaphic acid and tiliroside were observed under mildly acidic conditions (pH 6.00), while the lowest values were found under alkaline conditions (pH 9.00) (*Van De Waterbeemd et al., 1998*). The improved performance under mildly acidic conditions may be related to the chemical structure of the analytes, as the phenolic hydroxyl groups in Tiliroside and the carboxylic groups in euscaphic acid are more stable in such conditions, reducing degradation and enhancing solubility. Moreover, acidic environments can help break down cell walls and release bound polyphenols, improving extraction efficiency and potentially promoting pH-dependent active transport mechanisms. These findings indicate that mildly acidic conditions play a crucial role in optimizing the extraction and absorption of FRLE.

## CONCLUSIONS

In conclusion, based on the screening of the FRL extraction process, this study used the Caco-2 cell model to investigate the bidirectional transport characteristics of Euscaphic acid and tiliroside in FRLE, examining the effects of different concentrations, pH, temperature, time, and P-gp inhibitors on drug absorption. The optimal extraction conditions for FRL were identified as a solid-liquid ratio of 1:35 g/mL, 65% ethanol, and a reflux time of 135 min, maximizing the extraction yield, total polyphenol content, and euscaphic acid content. The transport of Euscaphic acid and tiliroside involves both active transport and passive diffusion, with good absorption within the tested concentration range. Mildly acidic conditions were found to enhance the transport and absorption of

these compounds, thereby increasing their bioavailability. These findings provide a foundation for the further development of FRL.

### Funding
The authors received no funding for this work.

### Competing Interests
The authors declare that they have no competing interests.

### Author Contributions
- Ning Wang conceived and designed the experiments, performed the experiments, analyzed the data, prepared figures and/or tables, authored or reviewed drafts of the article, and approved the final draft.
- Li Tian conceived and designed the experiments, performed the experiments, authored or reviewed drafts of the article, and approved the final draft.

### Data Availability
   The raw data is available in the Supplemental File.

### Supplemental Information
Supplemental information for this article can be found online at http://dx.doi.org/10.7717/peerj.18638#supplemental-information.

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
