# Peer review of "Study on the absorption characteristics of euscaphic acid and tiliroside in fruits of Rosa laxa Retz"

_PeerJ, doi:10.7717/peerj.18638_

## Round 0.1 · original submission · Major Revisions

Please carefully review the comments provided by the reviewers and address each query with a detailed response. As noted, the data presentation requires significant improvement. Kindly resolve these issues and incorporate the reviewers' suggestions in the revised version of the manuscript.

Reviewer 1 ·

Basic reporting

no comment

Experimental design

Lack of experimental method explanation for UPLC-TQ-MS and method explanation for different extraction conditions

Validity of the findings

Lack of experimental results and discussions on UPLC-TQ-MS and optimization of different extraction methods

Additional comments

1 The title and annotation of the figures should be placed below the corresponding figure.
2 Please replace Figure 5 and 6 with clearer images.
3 In the section 2.4, the information on the manufacturer of the resistance meter should be provided.

Reviewer 2 ·

Basic reporting

In this manuscript, Wang et al. optimized the extraction of bioactive compounds from Rosa laxa fruits using response surface methodology and UPLC-TQ-MS. They found that euscaphic acid and tiliroside, key components of the extract, are absorbed in the intestine primarily through passive diffusion, with P-glycoprotein-mediated basolateral efflux transport, suggesting potential pharmaceutical applications.
The topic of this work is well-aligned with the interests of PeerJ readers, and I recommend the manuscript for acceptance with minor revisions as outlined below:
1. In Figure 2, the size of the figure is too small, making it difficult for readers to interpret. Please consider enlarging it for better clarity.
2. The caption of Figure 3 appears to have formatting issues that need correction. The same applies to Table 4, which also has formatting problems.

Experimental design

no comment

Validity of the findings

no comment

·

Basic reporting

I would like to commend you on your research presented in the manuscript titled "Study on the Absorption Characteristics of Euscaphic acid and Tiliroside in fruits of Rosa laxa Retz." Your work addresses an important topic in the field of pharmacognosy and natural product chemistry, and it has the potential to contribute significantly to our understanding of these bioactive compounds.

However, I believe that more work needs to be done to enhance the overall quality of the manuscript. Specifically, improvements are needed in the design of the experiments, the presentation of the data, and the discussion of the findings in relation to the stated aims of your research.

Specific comments

Abstract
The extracted amount at optimized conditions is not mentioned in the abstract. It would be good to include it so that the readers can see immediately if the extraction method is efficient.

Introduction
1. What does it mean that RFL is an environmentally friendly plant?
2. Line 68 to line 74 are written to emphasize the potential of FRL. However, the suggestion is to use scientific language. EG. “Given the therapeutic potential of FRL components, a thorough evaluation and optimization of extraction techniques for these bioactive compounds is essential. The use of response surface methodology (RSM) is particularly advantageous in elucidating the complex interactions between evaluation parameters and individual factors (Zhang et al., 2022). This approach enables the efficient identification of optimal extraction conditions, accounting for the influence of multiple variables, and is especially relevant for studies focused on extraction processes in herbal medicine research (Addo et al., 2022).”
3. There is no mention of methods that have been used for extraction before. Does it mean this study is the first to optimize the extraction of euscaphic acid and tiliroside in FRL?

Figures
Figure 1. Show calibration curves, including the curve from the Folin-Ciocalteu method. Also, overlay UV-chromatogram showing the standard peaks and sample peaks. There is no need to have many separate plots.
Figure 2. The plots are small and not easy to see. It might be good to overlay plots as suggested for Figure 1.
Figure 3. It would be good to show the contour and the 3D plots of Euscaphic acid, Tiliroside and total polyphenol content separately. I firmly believe that the optimum condition for their extraction is different. A compositentive score does not communicate as a response. Unless there is an explanation of why it was chosen as a response, it must be clearly communicated in the method and discussed clearly.

Tables
Table 3 caption. Include what the response is.
Table 5. The data for Tiliroside is not included. Is it by mistake, or is there a reason for this? What is a Compositensive Score?
Table 7. The numbers in the table need to be aligned.

Experimental design

1. Line 88. Sampling time. The samples were collected in August 2019. Was there a particular reason for the time of sampling? Could the sampling time affect the amount of the analytes of interest present in the samples? Justification of the sampling time.
2. Line 108. What chromatographic system was used for this study?
3. Line 120. The precision, repeatability, stability and sample recovery rate were investigated and calculated. Please expand on how this was done.
4. Line 126. The writing style is inconsistent with the rest of the document, which is passive writing. The subsection needs to be written for consistency and to improve the grammar.
5. Line 135. Reflux extraction was chosen as an extraction method. Please clarify why this was chosen over other extraction methods, such as accelerated solvent extraction method, which is widely used to extract polyphenols from plant materials.
6. Line 136. Expand on how optimum conditions will be determined using Design Expert. Is there a function that predicts the optimum parameters?
7. Line 136 to 137. Expand on how the optimized extraction process is verified.
8. Line 161. Correct space “di luted” to “diluted”.

Validity of the findings

Results
1. Line 203 to line 206. While the chromatograms of the different compounds and the samples look impressive. Showing calibration curves would be more relevant since the results talk about the linearity of the curves. In addition, a single chromatogram of the standard mixture would be sufficient. Figure 1 shows that the peak resolution between Euscaphic acid and Tiliroside is very small. The question is, why was the resolution of the chromatographic separation not improved? Maybe it does not really matter because MS/MS is used. It would be good to mention it if it is the reason.
2. Line 207 to 211. It is not clear how recovery studies were done. It is not clear if it is extraction method recovery studies or analysis method recovery studies. Was the sample spiked? Please expand on this.
3. Line 214 to line 219. What does composite score mean? Does it mean the gravimetric yield? (What is the response being measured here?). It is not clear.
4. Line 221. Which model was used to fit the data? To show the model's validity, include the Lack of Fit tests. In addition, there are no results for Tiliroside. Is there a reason for the exclusion of this analyte?
5. Line 222. It is unclear if the observed effects have a negative or positive effect on the response (which is also not mentioned). In addition, please explain why the effects are observed in the way they are. It is valuable information for someone who wants to repeat the experiments.
6. Line 234 to line 239. What were the criteria set during optimization? Again, what is the relevance of the comprehensive score? Is it extraction comprehensiveness?
Discussion
1. Inconsistency in the spacing, e.g., Line 289 and Line 291
2. The discussion is only based on the absorption study. There is no further discussion on the extraction experiments and the analysis method.
3. Line 346 to line 349. Why are mild acidic conditions better? Does it have something to do with the structure of the analytes? Can you suggest an explanation for the observation?
Conclusions
1. The optimum extraction conditions may be good to mention here. The section should indicate whether the aims of the study were met.

Additional comments

No comments.

---

## Round 0.2 · accepted · Accept

As the reviewer noted, the authors should verify the accuracy of references, including formats, page numbers, and DOI numbers, in the final version of the manuscript.

Reviewer 1 ·

Basic reporting

no comment

Experimental design

no comment

Validity of the findings

no comment

Additional comments

The manuscript has improved a lot, however, I have noticed that some references formats are not standardized and lack volumes, page numbers, or DOI numbers. Therefore, please carefully check the reference information before Acceptance.

Reviewer 2 ·

Basic reporting

The authors addressed all my concerns and I recommend the acceptance of this manuscript.

Experimental design

no comment

Validity of the findings

no comment